# Lung Cancer and Air Quality in a Large Urban County in the United States

**DOI:** 10.3390/cancers16112146

**Published:** 2024-06-05

**Authors:** Hollis Hutchings, Qiong Zhang, Sue C. Grady, Jessica Cox, Andrew Popoff, Carl P. Wilson, Shangrui Zhu, Ikenna Okereke

**Affiliations:** 1Department of Surgery, Henry Ford Health, Detroit, MI 48202, USA; hollis.hutchings@gmail.com (H.H.); apopoff2@hfhs.org (A.P.); 2Department of Public Health Sciences, Henry Ford Health, Detroit, MI 48202, USA; qzhang2@hfhs.org (Q.Z.); cwilso28@hfhs.org (C.P.W.); 3Department of Geography, Environment and Spatial Sciences, Michigan State University, East Lansing, MI 48824, USA; gradys@msu.edu (S.C.G.); zhushang@msu.edu (S.Z.); 4School of Medicine, University of Texas Medical Branch, Galveston, TX 77555, USA; jrcox@utmb.edu

**Keywords:** air pollution, lung cancer, lung cancer screening, autoregressive distributed lag models (ARDLs), Michigan, USA

## Abstract

**Simple Summary:**

Lung cancer incidence varies geographically for several reasons, including environmental exposures. The incidence of lung cancer and corresponding air quality was examined over three decades in one large urban county in the United States. There were specific clusters of lung cancer cases that were found over that period. The most intense clusters corresponded to the areas of the county with the worst pollution. Sulfur dioxide, particulate matter < 10 μm, nitrogen dioxide, volatile organic compounds and ozone levels had significant relationships with lung cancer incidence. Different pollutants had different lag times between exposure level and lung cancer development.

**Abstract:**

Lung cancer is the leading cancer-related killer in the United States. The incidence varies geographically and may be affected by environmental pollutants. Our goal was to determine associations within time series for specific air pollutants and lung cancer cases over a 33-year period in Wayne County, Michigan, controlling for population change. Lung cancer data for Wayne County were queried from the Michigan Cancer Registry from 1985 to 2018. Air pollutant data were obtained from the United States Environmental Protection Agency from 1980 to 2018. Autoregressive distributed lag (ARDL) models were estimated to investigate time lags in years between specific air pollution levels and lung cancer development. A total of 58,866 cases of lung cancer were identified. The mean age was 67.8 years. Females accounted for 53 percent of all cases in 2018 compared to 44 percent in 1985. Three major clusters of lung cancer incidence were detected with the most intense clusters in downtown Detroit and the heavily industrialized downriver area. Sulfur dioxide (SO_2_) had the strongest statistically significant relationship with lung cancer, showing both short- and long-term effects (lag range, 1–15 years). Particulate matter (PM_2.5_) (lag range, 1–3 years) and nitrogen dioxide (NO_2_) (lag range, 2–4 years) had more immediate effects on lung cancer development compared to carbon monoxide (CO) (lag range, 5–6 years), hazardous air pollutants (HAPs) (lag range, 9 years) and lead (Pb) (lag range, 10–12 years), which had more long-term effects on lung cancer development. Areas with poor air quality may benefit from targeted interventions for lung cancer screening and reductions in environmental pollution.

## 1. Introduction

Lung cancer is the most common cause of cancer-related death in the United States [1]. In 2024, there were an estimated 234,580 new cases and 125,070 deaths due to lung cancer [2]. Lung cancer is also the leading cause of death among patients with cancer, accounting for about one in five of all cancer deaths in the U.S. [3]. One of the main reasons for the poor outcomes is that lung cancer is largely asymptomatic in the early stages of disease. As such, early detection is one of the strategies by which the medical community can reduce mortality from lung cancer. Although cigarette smoking remains the main risk factor for lung cancer development, the importance of environmental factors such as air pollution and poor air quality is becoming more recognized [4,5].

Air pollution appears to be a particularly important factor contributing to lung cancer, as evident by an estimated 25,000 annual deaths from lung cancer in non-smokers in the U.S. [6]. Other factors that contribute to the increased risk of developing lung cancer in non-smokers include secondhand smoke from cigarettes, occupational exposures and genetic susceptibility [7,8,9,10]. Several studies have indicated that exposure to air pollution, including fine particulates, nitrogen dioxide, sulfur dioxide, ozone, radon and asbestos, are associated with lung cancer [11,12,13]. Long-term exposure to air pollution may increase lung cancer risk through oxidative damage, which occurs via inflammatory injury and the production of reactive oxygen species [14,15]. Oxidative stress and inflammation can induce cellular damage and lead to the promotion of metaplastic cell survival.

Elevated levels of air pollutants in the U.S. can be found in the Midwest and Southeast regions of the country. In the Midwest, there are economically declining urban areas such as Detroit, Michigan [16]. Also referred to as the Rust Belt, these urban areas were historically manufacturing hubs for important industries, such as the automobile industry. With economic restructuring in the early 1970s, there was the movement of industries to the South and overseas. As residential areas were rezoned to industrial areas, low property values provided an incentive for new industries to move in. This movement led to a shift in air-borne lead-polluting industries from the suburbs to urban areas [17,18]. Several industrial plants and automobile factories are currently major sources of pollution and have led to seasonal trends in elevated air pollution levels [19].

The goal of this study was to investigate associations in the time series of specific air pollutants and lung cancer cases over three decades in Wayne County, Michigan. The spatial distribution and clusters of high lung cancer relative risk were mapped at the zip code level, and temporal trends in lung cancer incidence and air pollutants were graphed. Finally, the associations in the time series of specific air pollutants and lung cancer cases were explored for statistically significant time period(s) and lag time(s) in years, controlling for population changes, to inform future air pollutant and lung cancer studies. The findings from this study are also intended to inform the Michigan Department of Environment, Great Lakes and Energy, Wayne County Health Department, and highlight the need for improved lung cancer screening in areas of high air pollution.

## 2. Materials and Methods

### 2.1. Patients

Approval for the study was obtained from the Henry Ford Health Institutional Review Board prior to conducting the study (IRB-15286). The Michigan Cancer Surveillance Program (MCSP) is a state government-funded surveillance program that collects healthcare data for all patients in the state diagnosed with cancer at a participating treating institution. Every patient who developed lung cancer in the state from 1985 to 2018 was included in the database.

### 2.2. Study Area

The study was conducted in the city of Detroit and Wayne County, Michigan. It consists of urban and suburban regions. Detroit is the largest city in Wayne County. Wayne County is the most populous county in the state of Michigan, with a population of 1.8 million people in 2020. In the county, 48 percent of people are White Americans, 38 percent are Black Americans, 7 percent are Hispanic Americans and 7 percent are other races [20].

### 2.3. Study Design and Settings

A retrospective cross-sectional study design was implemented utilizing the MSCP database queried from 1985 to 2018 for all lung cancer cases in Wayne County using the ‘county of diagnosis’ (FIPS code = 163) (n = 58,866). Patient demographics (sex, age and race), stage of cancer (extent of disease) and survival were recorded overall and at the patients’ zip code level. Stage was categorized in the database as early, regional and distant. Regional or distant disease was considered to be advanced stage. Age was considered to be the patient’s age at the time of diagnosis. Although smoking history was a variable in the MCSP database, roughly 85 percent of patients had missing data for this variable. Furthermore, no zip-code-level smoking rates were available for Wayne County during the study period.

### 2.4. Air Quality

Daily air quality–pollution data from 1980 to 2021 were obtained from the United States Environmental Protection Agency (EPA) AirData website for Michigan. A total of nine pollutants and their levels were collected from monitoring stations, including carbon monoxide (CO), sulfur dioxide (SO_2_), nitrogen dioxide (NO_2_), ozone (O_3_), particulate matter < 2.5 microns (PM_2.5_), particulate matter < 10 microns (PM_10_), lead (Pb), hazardous air pollutants (HAPs) and volatile organic compounds (VOCs). CO and O_3_ were measured in parts per million. NO_2_ and SO_2_ were measured in parts per billion. Lead, HAP, PM_2.5_ and PM_10_ were measured in micrograms per cubic meter. VOCs were measured in parts per billion carbon. While the levels of air pollutants were available at each air monitoring station, an empirical Bayesian Kriging model was used to interpolate (estimate) values between monitoring stations, thereby resulting in a continuous surface of annual average air pollution estimates for each air pollutant in Wayne County [21].

### 2.5. Statistical Analysis

Firstly, descriptive statistics of the lung cancer patients were generated. Secondly, the cumulative relative risk (RR) of lung cancer incidence across the study time period was calculated at the zip code level using the 2010 American Community Survey 2018 5-year population estimates in SaTScan (v9.7) software [22]. The observed number of lung cancer cases was divided by the expected number estimated using a Poisson model to obtain the RR for each zip code and also to identify areas where the observed numbers were significantly higher than expected. These areas in which observed incidence was significantly higher were referred to as lung cancer clusters. Lung cancer clusters were optimized using the Gini coefficient to detect smaller non-overlapping clusters instead of overly large clusters in the study area [23]. The Gini coefficient is a measure to describe how closely the expected number of cases is to the observed number of cases in different areas. As such, within one overly large cluster, there may be additional smaller clusters that a high Gini coefficient would detect. The temporal patterns of lung cancer incidence and pollutant levels were also presented graphically over the study time period.

Thirdly, the time series of specific air pollutants and lung cancer cases were investigated by estimating autoregressive distributed lag (ARDL) models. The ARDL model utilizes a linear regression model to find significant time period(s) and lag time(s) in years within those time periods between an air pollution level (independent variable) and lung cancer cases (dependent variable), controlling for trends in varying population size. For this study, a maximum of 15 years’ lag or less was studied to assess each air pollutant’s relationship with lung cancer cases. Using this method, a “significant” lag would represent the number of years from a pollutant’s levels and the magnitude of lung cancer cases [24].

## 3. Results

### 3.1. Lung Cancer Statistics

A total of 58,866 new cases of lung cancer were identified in Wayne County. The demographics are listed in Table 1. The mean age was 67.8 years. Thirty-seven percent of patients identified as African American. The largest proportion of lung cancer diagnoses presented in advanced stages, with 20 percent of patients presenting with regional metastases and 48 percent of patients presenting with distant metastases.

More men (56.8%) than women were diagnosed with lung cancer. The gender differences narrowed over time, however. In 1985, men were 2.2 times more likely to develop lung cancer than women, with men representing 66 percent of all lung cancers in 1985. By 2018, the incidences of lung cancer for men and women were relatively similar, with women representing 53 percent of all lung cancers (Figure 1).

### 3.2. Spatial Distribution

Figure 2 shows the spatial distribution of lung cancer relative risk (RR) at the zip code level in Wayne County (RR range, 0.17 to 2.1). There were three significant clusters of lung cancer RR found (a) in downtown Detroit (RR range, 0.86–2.02), (b) in the downriver Detroit area (RR range, 1.11–2.02) and (c) the western suburbs of Detroit (RR range, 1.11–1.47).

### 3.3. Temporal Trends in Lung Cancer Incidence and Air Quality

Figure 3 shows the temporal trends of lung cancer incidence across the study time period. There was a declining trend beginning in the mid-1990s and particularly after 2005.

Figure 4 shows the temporal trends in the nine air pollutants studied. While most pollutant levels decreased over time, the levels of PM_2.5_ increased in Wayne County in the last 5 years.

### 3.4. Associations between Air Pollutants and Lung Cancer

The results of the ARDL model also showed that multiple air pollutants had a significant impact on the trend of lung cancer incidence after controlling for population variability. Figure 5 provides the results of comparative analyses of those air pollutants, time periods and lag time(s) in years in which specific air pollutants had a statistically significant effect on lung cancer cases. All air pollutants demonstrated a significant relationship with lung cancer cases between the time periods within which air pollution data were collected (1985–2018). Within these time periods, the pollutant with the widest range in significant lag times was SO_2_ (lag range 1–15 years). There was also a large range in lags for PM_10_ (lag range 2–8 years) and O_3_ (lag range 1–8 years). There were more immediate effects of PM_2.5_ (lag range 1–3 years) and NO_2_ (lag range 2–4 years). In contrast, there were more delayed effects of other air pollutants on lung cancer development, including HAP (lag range 9 years) and Pb (lag range 10–12 years). VOCs were not a good fit in the ARDL models due to the lack of temporal changes, as the VOC pollutant levels were quite stable from 1996 to 2018 (Figure 4). As such, a generalized linear regression model was applied to study the concurrent relationship between VOCs and lung cancer cases. The results showed a significant relationship between VOCs and lung cancer from 1991 to 2018 (*p* < 0.01).

## 4. Discussion

This study found that air pollution had a significant association with lung cancer development in Wayne County, Michigan, with varying lag times. These relationships have been seen in multiple studies that have linked air quality to the development of lung cancer, although these studies did not provide evidence of lag time(s) between potential exposure to air pollution levels and lung cancer [25,26,27,28,29]. Ambient particulate matter exposure was previously demonstrated to contribute to lung cancer development in males and older patients as the cumulative dose increased [30]. Additionally, the ESCAPE study conducted in Europe demonstrated a statistically significant relationship between PM_10_ and PM_2.5_ exposure and the development of lung disease, in addition to distance to heavy road traffic areas [31]. Our study is novel in that it shows that different pollutants have different lag times between pollutant levels and lung cancer. Previous work has demonstrated how cumulative exposure to highly pollutant regions can increase risk for the development of lung cancer. Further work is needed to fully understand the significance of short lag times on lung cancer incidence.

When evaluating the incidence of lung cancer in Wayne County during our study period, we observed that men initially presented with higher rates of lung cancer than women. By the end of our study period, however, men and women developed lung cancer at nearly equal rates. This is likely due to multiple factors, including the overall decreasing rate of smoking in the United States. While smoking has decreased overall, the rate of smoking has decreased more dramatically in men than in women. Overall, the rates of lung cancer have declined over time in the United States. In 1999, the age-adjusted new lung cancer diagnosis rate was 70.7 per 100,000 people. In 2020, that rate dropped to 47.1 per 100,000 people [32]. The falling rates of lung cancer over time, in addition to decreasing smoking rates, likely contributes to the trend we see in our lung cancer dataset.

Although our study was only performed in Wayne County, it is expected that these results should be applicable to other urban areas with similar levels of air pollution. By identifying high-risk areas, hospitals and public health groups can better engage with the most at-risk populations to address disease prevention.

The rise in industrialization, traffic density and population density has led to changes in air quality over the last 100 years [29,33]. It is important for the medical community to understand these relationships as we seek to lower mortality from lung cancer. Currently, most caregivers ask for patient information such as age, smoking history and occupational history to determine risk factors for lung cancer development. In the future, it will be important to understand a patient’s residential history as well. If it can be determined that a patient has lived in an area of poor air quality for a prolonged period, increased surveillance may be warranted.

In our study, SO_2_ appeared to have the strongest lag relationships with lung cancer cases. SO_2_ pollution has been linked to respiratory diseases in multiple studies [34,35,36]. Most studies, however, have focused on benign disease. Our study is among the very few to show strong immediate and delayed relationships between SO_2_ and lung cancer. Our study was aided by two important factors. Firstly, we utilized pollution data from the EPA over a very long time period. We hypothesized that any relationship between pollution and cancer would potentially take decades to be revealed. Secondly, Michigan has had a comprehensive cancer database since 1985. In this database, every patient who developed lung cancer was recorded. Utilizing these two factors, we were able to show a significant relationship between SO_2_ and lung cancer time series. As a major manufacturing hub since the 1880s, Detroit and Wayne County have had some of the highest SO_2_ levels in the country for decades. In 2013, Detroit was one of several cities that failed to meet the 2010 air quality standard set by the Clean Air Act. As a result, the EPA in 2022 released a Federal Implementation Plan to improve air quality in this area [37]. We feel that other cities across the country and world with high SO_2_ levels should implement similar plans in their regions. Lower SO_2_ levels should reduce incidence of both benign and malignant lung-related illnesses.

We also saw that environmental VOC levels correlated with lung cancer incidence in our study. There is significant ongoing research about the use of inhaled VOC levels to encourage screening or detect lung cancer at early stages [38]. Although the compounds and assays are different, it is worthwhile to study any associations between exhaled lung biomarker assays and environmental pollutant levels. By identifying areas which are high risk for the development of disease, interventions like lung cancer screening and air quality improvement can be targeted and potentially have greater yield. By linking local businesses, government agencies, medical institutions, and community partners, we can use the information gained in this study to intervene at the patient level to reduce mortality from lung cancer.

This study has limitations. One major incomplete variable was individual or zip-code-level smoking history. It is possible that more industrialized areas have residents who are more likely to smoke. In addition, there have been changes in smoking rates over time. From 1965 to 2006, the rate of smoking in adults nationally decreased from 44 percent to 21 percent [39]. Unfortunately, long-term data do not exist concerning smoking rates at a local level. Despite this limitation, it is still important to investigate potential associations between environmental pollutants and lung cancer incidence. Additionally, the only publicly available air quality data were over the 30-year study period from the EPA, and the density of air quality sensors was relatively low in Wayne County, Michigan. The length of data availability limited the number of time lags that could be tested from a statistical perspective. Lung cancers and other solid tumors generally have latency periods of 20 to 30 years [40]. Furthermore, establishing a higher density of air sensors will allow for better data collection and the ability to make precise measurements. The establishment of a local air quality network using consumer-level instruments is one method to increase data collection and better understand the relationship between air quality and lung cancer. Finally, this study was unable to capture any migration patterns in patients. The large number of lung cancer cases, however, likely mitigates any migration bias to a large degree.

## 5. Conclusions

Our study revealed associations between air pollutants and lung cancer incidence over a 33-year period. Lag times between pollutant levels and lung cancer cases ranged and varied for different pollutants. Medical institutions can utilize this information by querying patients about their residential histories in addition to other information such as smoking and occupational exposure. In addition, the business community can work with government officials to improve air quality over time.

## Figures and Tables

**Figure 1 cancers-16-02146-f001:**
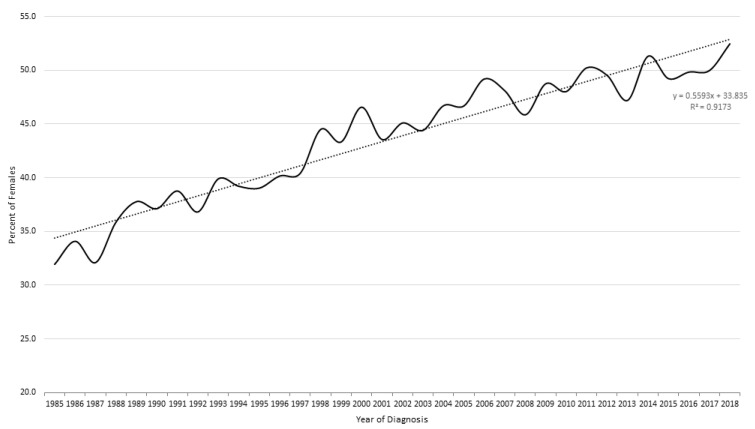
Gender ratio of lung cancer cases over time.

**Figure 2 cancers-16-02146-f002:**
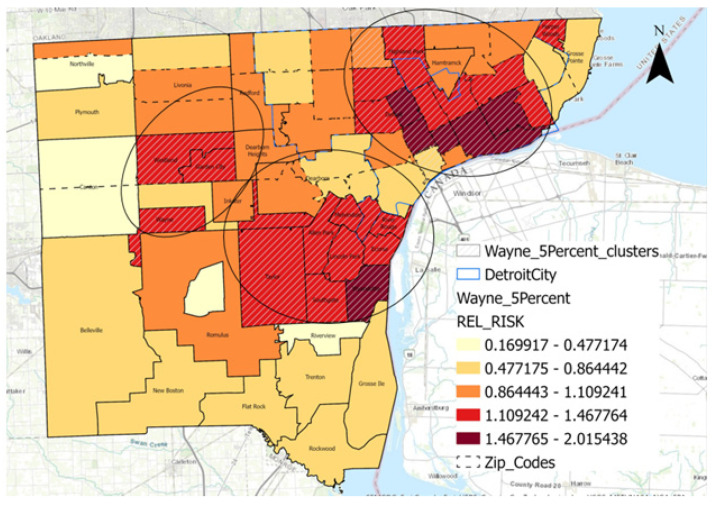
Geographic distribution of lung camcer cases over time.

**Figure 3 cancers-16-02146-f003:**
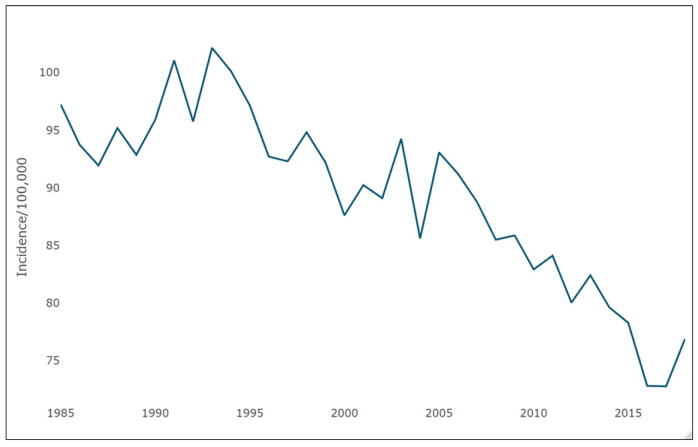
Trends in incidence of lung cancer over time.

**Figure 4 cancers-16-02146-f004:**
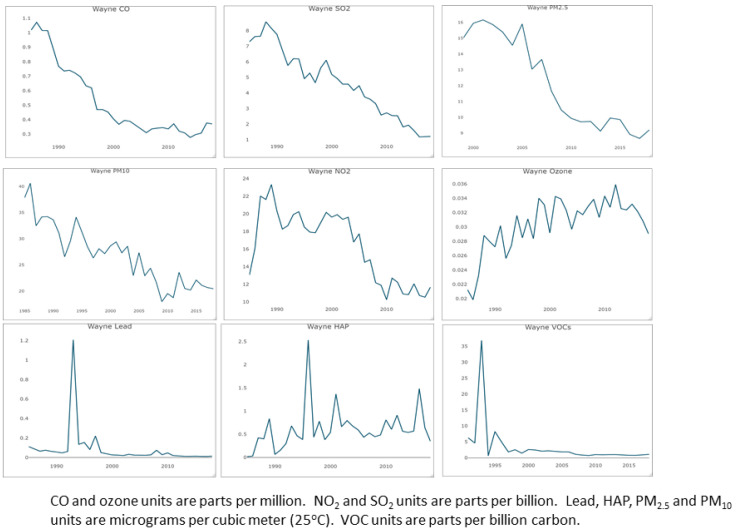
Pollutant levels over time.

**Figure 5 cancers-16-02146-f005:**
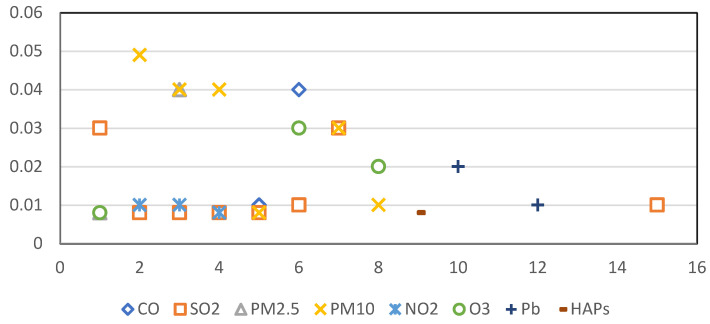
Temporal trends in lag time for the nine air pollutants studied.

**Table 1 cancers-16-02146-t001:** Lung cancer patient demographics in Wayne County, Michigan, from 1985 to 2018.

N	58,866
Age, years (mean)	67.8 ± 11.14
Men	56.8%
Race	
Caucasian	62.0%
African American	37.2%
Other	0.8%
Extent of disease	
Localized	10,197 (17.3%)
Regional	11,761 (20.0%)
Distant	28,494 (48.4%)
Unknown	8404 (14.3%)

## Data Availability

Data were obtained from the Michigan Cancer Surveillance Program and the United States Environmental Protection Agency. Data from the Michigan Cancer Surveillance Program are unable to be shared secondary to a binding agreement through this program. All data are available directly through the Michigan Cancer Surveillance Program upon request and the completion of application for data. Data from the United States Environmental Protection Agency are also available upon request.

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
