# Peer review of "Lung Cancer and Air Quality in a Large Urban County in the United States"

_cancers, 2024, doi:10.3390/cancers16112146_

Round 1

Reviewer 1 Report (Previous Reviewer 1)

Comments and Suggestions for Authors

The paper is much improved and now reads reasonably clearly and makes an important contribution. I have two small but substantive concerns where the paper can be improved, and a few minor editorial points.

In the first point under 3.4, I don’t understand what is being said about VOCs and lung cancer. Why is the model different from that used with other pollutants, and what does it mean that results showed a significant relationship from 1991-2018?

Table 2 is much improved and now makes more sense. I have a question, however. Am I correct that for each pollutant, there was a result calculated for each annual lag between 1985 and 2018, for a total of 34 results? And then in Table 2, you present only the years in which the p-value was < 0.05? To take a specific case, am I correct that for SO2, the p-values for lags 8 through 14 years were > 0.05? If this is the case, then I think a much better way to show the results in Table 2 would be with a graph for each pollutant, with the p-value for the lag on the y axis and the range of lags on x. With so many p-values, one should not really think of each of these as an individual test of a specific null hypothesis, because of the multiple comparisons problem. Instead, you could simply show the full set of lags and the graph provides the full set of results. Your main points will be more easily understood in this format.

Some additional editorial suggestions (line numbers are from the MS Word document, with Track Changes turned off)

Simple Summary, line 18: either PM10 or particulate matter < 10 micrometers

Abstract, lag ranges for PM2.5 and Pb should be written 1 – 3 and 10 – 12 respectively.

Introduction, references 5 and 10 – I don’t see what these have to do with the role of environmental factors in lung cancer.

Page 3, lines 95-96: minor point, but if you’re going to include these percents, you somehow have to get them to sum to 100 – probably with “other”.

Page 3, line 113:  particulate matter < 10 microns (PM10)

Page 4, line 131, Please provide a reference (not to a software manual) for the Gini coefficient.

Page 7, Table 2: I suggested a set of figures instead, but if you include this table, please use consistently 2 digits for p-values. Going to 3 digits for .049 gives the impression that you are chasing statistical significance.

Page 9, lines 288 – 297: this discussion of VOC is improved. However, I think you are still mixing inhaled and exhaled VOCs in the citations 38 – 42. References 38, 41 and 42 are about air pollution, while 39 and 40 concern VOCs as biomarkers of lung injury.

Page 9, lines 308, 309: I think this sentence needs a citation.

Author Response

Reviewer 2 Report (Previous Reviewer 3)

Comments and Suggestions for Authors

Dear Authors,

Thank you for responding to my previous comments. Your answers are satisfactory and article is suitable for publication.

Good luck with the next step of the processing of your paper.

Author Response

This manuscript is a resubmission of an earlier submission. The following is a list of the peer review reports and author responses from that submission.

Round 1

Reviewer 1 Report

Comments and Suggestions for Authors

This is a novel study with some potentially important findings. The basic data, methods and analyses appear to be sound. There are a number of improvements to the text that would substantially improve the paper.

 Methods. The analytical methods are not sufficiently described for readers with a background in cancer epidemiology. For example, page 3, lines 103-105: “For each limit a hierarchical non-overlapping set of clusters were detected and a Gini Index was used to determine the optimal cluster reporting sizes for the Poisson model spatial scan statistic [15].” This is not clear. Please explain what this method is doing. What is the Gini Index? What is a spatial scan statistic? These are not common methods in cancer epidemiology. Similarly, on page 4, lines 122-123, please explain. There is not even a reference, let alone a definition, for the Phillips-Perron test. Further on in the same paragraph, this is not clear: “The sensitivity analysis was conducted on 15 years’ lag. Additionally, bounds testing was used to measure the significance of cointegration between air pollution and lung cancer by using the ardlBound function in “dLagM” package.” This is not clear.

 Results. In general, the results are under-described.

 Figure 1 is quite impressive. But the authors might help the reader understand this linear trend in the sex ratio. Why has it occurred? What does it mean for the findings of the paper? I suspect that it is driven be several underlying trends, including the decline in smoking in men, and perhaps also shifts in the age distributions of men and women, and the sex ratio of the county population. What is the trend in the rates of lung cancer in men and women over time?

 On page 5, the findings on the Spatial distribution. We are not given any data with which we can evaluate the optimal scanning window bandwidth. How sensitive were these cluster findings to the choice of bandwidth?

 On page 7, lines 163 – 173, these findings are not at all clear. Please explain why an ARDL model must have I(1) as a dependent variable, but can process both I(0) and I(1) as independent variables, and what this means? What is the meaning and relevance of stationarity in these data? We are not given data with which to evaluate the optimality of the lags. These seem to be among the most important findings, but we have no way to evaluate them. How much more important was one lag versus another? Also, we are not told the range of possible lags that were evaluated.

 The findings on SO2 appear to be among the most important, according to the authors, but the reader lacks evidence with which to evaluate this. It is not clear what is meant by: “The bounds testing showed that lung cancer and SO2 correlated with at least 1 co-integration (F-statistic = 6.44), showing that SO2 and lung cancer shared similar long-term temporal trends.”

 Table 2 will not be interpretable to the reader with a background in cancer epidemiology.

 Discussion.

 The paper reads as if the authors are not aware of the large literature on induction time and latency for environmental carcinogens. It is potentially quite useful to use empirical methods such as these to measure latencies, but then these findings should be set in the context of what is already known. It is well-accepted that cancer initiators have induction times of decades, while promoters, or late stage carcinogens, may act with only a few years’ latency. The reader should be provided with some of this context. Also, the citations on air pollutants and lung cancer could be improved. It is surprising, for example, that the IARC monograph 105 on outdoor air pollution is not cited. Citation 22 concerns COPD, not lung cancer. This sentence seems to suggest that the authors are not aware of the extensive past literature: “Interestingly, our analyses showed that different pollutants have different lag times and levels of correlation to lung cancer incidence.” Yes, but what does this suggest, and how does it confirm or contradict what is already known?

Sulfur dioxide is not known to be a human carcinogen. Of course, perhaps this study is finding something new, contradicting past studies. Fine, but please note this for the reader.

Reviewer 2 Report

Comments and Suggestions for Authors

Thank you for the opportunity to review this manuscript. The manuscript “Lung cancer and air quality in a large urban county in the United States” presents an important topic. Considering broader context of presented studies and further impact of the published papers, there are some corrections/suggestions that I have pointed out; suggested to be addressed.

1.      Abstract, Line 33: “Varying lag times”, It would be better to state the range of varying lag times for understanding.

2.      Abstract, Line 29-32: I would suggest adding analysis values of the results stated. It is difficult to understand the impact of results without values.

3.      The authors have stated that the mean age of the study participants was 67.8 years. However, it is not clear whether the mean age was at time of inclusion, diagnosis, study?

4.      Table 2: “Optimal lag and significant lag” are not clear. It would be good to mention it with the unit considered in the study.

5.      The authors have defined lag times. However, it is still not clear how these lag time were considered and calculated in the current study. It will be good to elaborate them a little more in the methodology section.

6.      Similarly, it would be good to mention further detail and comparatives about lag times finding of the study in the discussion section.

Few minor corrections:

1.      I would suggest replacing “Air Quality” and “Lung cancer” in key words with suitable similar words. As these words have already been used in the title of the manuscript.

2.      Line 225-226. The words “in our study” have been repeated twice in the same sentence.

Reviewer 3 Report

Comments and Suggestions for Authors

Review for the Manuscript ID: 2979429

Authors investigated important issue of lung cancer incidence in relation with air pollution over four decades in US. The topic and the results are important and interesting for the scientific audience. However, the reviewer suggest corrections for manuscript improvement.  Below are the comments of reviewer to the authors of the paper:

Introduction:

1.In the introduction, it is worth to include the information on the level of air pollution in the US compared to other highly developed countries. In addition, it is recommended to describe the current trends in air pollution in the whole country, and in the various states of US. It would be interesting for reader to know, if in various parts of US the air pollution differs. Maybe authors could mention the sources of pollutants.

2. A description is needed on the importance of the urban environment on the development of lung cancer.

3. To justify the study, the authors should add, what research on this topic has been conducted so far in the US, especially in the urban settings. Please, highlight the gap of this research in comparison with previous studies.

4. In line 53, there is mentioned “several studies”, but there is shown only one reference (10). Please, add more references.

Materials and Methods:

5.In this section, it need to add new paragraphs “Study design and settings” and “Study area”. It these parts authors will describe significant characteristics of the study.

6. There is a little information on how lung cancer data links with environmental pollution. Please, describe it in detail.

7. Please, describe what units were used in the study for the 9 types of air pollution.

8.In line 104, the authors mentioned “ Gini Index”, but did not describe, what it is.

9. There is not mentioned p-value in the methods.

Results:

10. In the description of the results, the values should be added (139-140 and 158-160).

11. In the title of Table 2, please, add time delay. Under Table 2, please, add abbreviations for the nine pollutants.

Discussion:

12. Information regarding future research should be placed in one place (from lines 208 and 243). Similarly, the limitations should be in one place as well.

Reviewers’ recommendation: Accept after major revisions.

Round 2

Reviewer 1 Report

Comments and Suggestions for Authors

I am concerned that the paper remains unclear in many important aspects, despite detailed suggestions for how it could be improved. Also, it appears that the revisions have been inserted by someone whose first language is not English, and this has hurt the clarity of the paper.

Some additional comments and suggestions follow.

Introduction. On page 2, lines 60 – 70, a paragraph has been added about air pollution that is both too broad and too narrow. The role of developed countries in global air pollution is interesting background, but not very helpful for this study. The discussion of the individual pollutants is misleading – it is true that VOCs are found in aerosol sprays, cleaners and insect repellants, but these are unlikely to be important sources of the geographic variability at the census tract level that is being analyzed here. Similarly, construction sites, wildfires and gravel pits are unlikely to be at the top of the list of the sources of particulate matter that drives the geographic variability observed in Wayne County. There are also no references on the sources of air pollutants.

 Methods. In this revised manuscript, the analytical methods remain insufficiently described for readers with a background in cancer epidemiology.

Some examples:

p. 4, lines 133-136. These sentences have been added, but I am not any clearer on what the methods accomplish, and the English is poor.

p. 4, line 137. Cointegration has not been defined.

p. 4, lines 144-147. These sentences on methods for choosing the lag remain unclear. What does significant mean here? More precisely: what is the null hypothesis that the p-value of the significance test evaluates?

p. 4, lines 153 – 156. This jargon is not defined. What is I()?

Results. The results remain under-described.

p. 6, line 197. I thank the reviewers for including results on the Gini statistic in their response to comments. But the readers of the paper will still be in the dark. Also, what the graph of Gini statistics shows is that the Gini coefficient for the 5% window is only marginally better than the 30% window. I think the reader might want to know this. Also, the Gini statistic is never defined.

p. 7, Figure 3. These are very large decreases in air pollution levels. I would think they would warrant some comment. How would these important trends impact the clustering findings?

p. 8, lines 217-227. This paragraph is unchanged.  As in my previous comments  - these findings are not at all clear. Please explain why an ARDL model must have I(1) as a dependent variable, but can process both I(0) and I(1) as independent variables, and what this means? What is the meaning and relevance of stationarity in these data? We are not given data with which to evaluate the optimality of the lags. These seem to be among the most important findings, but we have no way to evaluate them. How much more important was one lag versus another?

Discussion

p. 10, line 257. Lags of 1 to 8 years were investigated. But it is well-accepted that for lung cancers and other solid tumors, minimum latencies of 20 to 30 years are to be expected. At a minimum, this limitation of your data should be noted.

p. 10, lines 276 – 277. You say that air quality is worsening, but Figure 4 shows the opposite?

p. 11, lines 303 – 305. I think you mean exhaled VOC here, not inhaled? It’s interesting to think about the possible relationships between exhaled lung biomarkers of tumorigenesis and air pollution, but I think the VOCs being measured are very different chemical compounds?

Comments on the Quality of English Language

The original draft was in clear, correct English. Revisions for the second draft have numerous mistakes.

Reviewer 3 Report

Comments and Suggestions for Authors

Dear Authors,

After checking the manuscript, I believe that the manuscript has improved considerably now, but I can see that, unfortunately, some of my comments have not been addressed. I would request to respond to the following points from Review 1:

1. It is recommended to describe the current trends in air pollution in the whole country, and in the various states of US.

2.A description is needed on the importance of the urban environment on the development of lung cancer.

5.In section Materials and Methods, it need to add new paragraphs “Study design and settings” and “Study area”. It these parts authors will describe significant characteristics of the study.

6. In section Materials and Methods is a little information on how lung cancer data links with environmental pollution. Please, describe it in detail.

For the correcting their manuscript it may be helpful for authors to have a look at the similar articles published in the MDPI, for instance,

https://doi.org/10.3390/cancers16061189

https://doi.org/10.3390/ijerph20095707

and STROBE Checklists https://www.strobe-statement.org/checklists/

I expect to read a proper revision.
